# Emerging Role of NLRP3 Inflammasome/Pyroptosis in Huntington’s Disease

**DOI:** 10.3390/ijms23158363

**Published:** 2022-07-28

**Authors:** Emanuela Paldino, Francesca Romana Fusco

**Affiliations:** 1Laboratory of Neuroanatomy, IRRCS Santa Lucia, Via del Fosso di Fiorano 64, 00143 Rome, Italy; f.fusco@hsantalucia.it; 2Department of Systems Medicine, University of Rome Tor Vergata, 00143 Rome, Italy

**Keywords:** NLRP3 inflammasome, neuroinflammation, Huntington’s disease

## Abstract

Huntington’s disease (HD) is a neurodegenerative disease characterized by several symptoms encompassing movement, cognition, and behavior. The mutation of the *IT15* gene encoding for the huntingtin protein is the cause of HD. Mutant huntingtin interacts with and impairs the function of several transcription factors involved in neuronal survival. Although many mechanisms determining neuronal death have been described over the years, the significant role of inflammation has gained momentum in the last decade. Drugs targeting the elements that orchestrate inflammation have been considered powerful tools to treat HD. In this review, we will describe the data supporting inflammasome and NLRP3 as a target of therapeutics to fight HD, deepening the possible mechanisms of action underlying these effects.

## 1. Introduction

Huntington’s disease (HD) is an autosomal dominant neurodegenerative disorder characterized by a specific phenotype that includes chorea, dystonia, motor impairment, cognitive decline, behavioral, and psychiatric disorder [1,2]. The mutation occurs in the *IT15* gene, encoding for the huntingtin protein (The Huntington’s Disease Collaborative Research Group, 1993). Mutated huntingtin has a CAG repeat expansion which exceeds the normal range of about 10–35 [3]. The severity of the disease is related to the number of CAG repeats, determining the variation in the age of onset. One of the most described features of mutated protein is its capacity to accumulate and aggregate [4,5] forming the intranuclear and intracytoplasmic neuronal inclusions (NIIs) which are now considered a pathological hallmark of the disease [6,7,8,9].

Early damage is most evident in the striatal part of the basal ganglia in HD. In particular, the spiny projection neurons, which constitute about 95% of the striatum, degenerate massively in HD [10].

One of the mechanisms underlying the vulnerability of striatum in HD is explained by the fact that these neurons do not synthetize enough brain-derived neurotrophic factor (BDNF) [11]. BDNF is very important for the survival of mature neurons in the striatum. Striatal BDNF depends on the cortex for its synthesis and release, as it is synthesized by cortical neurons and released in the striatum through cortico-striatal anterograde transport. This microtubule-based transport depends on huntingtin and is altered in HD [12].

Moreover, the cAMP response element-binding protein (CREB), is an important transcription factor whose function is impaired by mutated huntingtin. In fact, cAMP levels are decreased in cerebrospinal fluid of HD patients [13] and transcription of CREB-regulated genes is reduced in the R6/2 transgenic mouse model of HD [14,15,16]. Drugs targeting CREB loss of function and BDNF decrease have been considered powerful tools to treat HD [17,18].

The role of inflammation has gained, in the last decade, an increasingly important role in the context of neurodegenerative diseases. Inflammation is a physiological response aimed at repairing damaged tissue in several different conditions. The sophisticated machinery that regulates the inflammatory process is activated to initiate the healing mechanism, ensuring the protection of the organism [19,20]. A very important role in the central nervous system is played by the microglia. Activated microglia can produce several dangerous molecules, such as prostaglandin E2 (PGE2), nitric oxide (NO), and cytokines such as tumor necrosis factor-a (TNF-a), interleukin-1 b (IL-1 b), and interleukin 6 (IL-6), that participate in the pathological processes through the activation of nuclear factor-k B (NF-k B), thereby leading to the neurodegeneration [21,22,23]. The inhibition of microglial malignant phenotype, through the reduction in pro-inflammatory factors, is one of the aspects on which the attention is being focused to treat or alleviate the most severe symptoms of neurodegenerative disorders such as HD.

## 2. Neuronal Cell Death in Huntington’s Disease

In physiological conditions, cell death is a highly regulated homeostatic mechanism and is considered crucial to maintaining tissues, organ size, and function [24]. On the other hand, in pathological conditions, cell death is the final solution only when multiple escaping-stress mechanisms failed, leading to the inability of a cell to recover its physiological status [25]. Neurodegenerative disorders are characterized by cytotoxic events that lead to an increased number of reactive oxygen species (ROS) formation, inflammation, synaptic dysfunction, excitotoxicity, impaired protein degradation system, endoplasmic reticulum (ER) stress, DNA fragmentation, and mitochondrial dysfunction [26]. In HD, mutant huntingtin is involved in neuronal dysfunction and death leading to the activation of different molecular pathways.

### 2.1. Necrosis

Necrosis is observed in many pathological conditions, especially in neurodegenerative disorders that are characterized by glutamate excitotoxicity including Parkinson’s disease and HD [27]. In the seventies, Olney and coworkers (1974) [28] suggested a model where the “necrotizing effect is, in essence, an exaggeration of the excitatory effect” and conceived the term “excitotoxic amino acids”. Excitotoxicity was first implicated in the 1980s in the ischemic brain damage, with a general recognition of glutamate as the main player of the mechanism of neuronal death [29]. Many studies, over about two decades, aimed at counteracting excitotoxic necrosis by administering glutamate receptor antagonists. By the early 2000s, compounds that were used had failed in multiple clinical trials [30,31,32] and research in the field of excitotoxicity lost its momentum.

### 2.2. Apoptosis

Apoptosis is a classic form of programmed cell death, and it is characterized by specific cytomorphological features, such as cell rounding up, pseudopod shrinkage, decrease in cellular volume, chromatin condensation, and nuclear fragmentation. These features are accompanied by little or no ultrastructural deformations of organelles in the cytoplasm, followed by plasma membrane blebbing and ingestion by phagocytes [33]. Many stimuli could initiate apoptotic cell death, including oxidative and metabolic stress, excitotoxicity, neurotrophic factor withdrawal, and toxins. The signals activating cell death could be extrinsic or intrinsic: the extrinsic apoptotic pathway is triggered by the specific binding of tumor necrosis factor (TNF)-family death receptors to the cell surface; on the contrary, under stress conditions, the intrinsic apoptotic pathway starts from an imbalance between pro-apoptotic Bcl2 family members (Bax, Bak, etc.), and anti-apoptotic members toward pro-apoptotic ones into the mitochondria [33]. In mammalian cell apoptosis, proteolytic enzymes called caspases play a key role, they are strictly involved both in extrinsic and intrinsic apoptotic pathways [34]. In order to demonstrate the occurrence of apoptosis in HD brains, it was important to evaluate the increase in caspase-1 and caspase-8 activity [35,36]. Additionally, caspase-9 is seen to be able to regulate caspase-6 activity which is strictly dependent on CAG repeat size in human HD brains [37]. Our group showed the expression of apoptosis and its modulation in the quinolinic acid lesion model of HD [38]. Consequently, studies on the role of mitochondria in disease progression showed an increased mitochondrial membrane depolarization in HD striatal projection neurons [39]. Moreover, mHTT is able to reduce calcium buffering capability and a loss of mitochondrial membrane potential [39]. After cytochrome C is released from mitochondria, mHTT is cleaved by caspases supporting its translocation into the nucleus in order to interact with different transcription factors, for example, p53, which regulates pro-apoptotic factors expression determining mitochondrial disruption [40,41]. These investigations suggest that mHTT could be involved in mitochondrial dysfunctions and abnormalities demonstrating its key role in the progression of HD pathogenesis.

Our group showed that PARP inhibition by INO-1001 exerted a beneficial effect on the R6/2 mouse model of HD in terms of survival, neurological impairment, and neuroprotection. PARP inhibition promoted striatal neuron survival in the animal model of HD, R6/2 and it was associated with an upregulation of CREB and CREB phosphorylation promotes an increase in BDNF. PARP-1 inhibition reduced apoptosis, as confirmed by our study, and was associated with a modulation of BAX and BCL-2 expression. We speculated that the inhibitory effect on apoptosis resulted in increased pCREB levels, which in turn upregulated downstream survival factors such as BDNF [42].

### 2.3. Necroptosis

The necrotic death pathway has also emerged as a type of active programmed cell death. Among these pathways, the best-characterized one is termed “necroptosis” [43,44]. In the past, necrotic cell death has been considered an event without genetic determinants since it is not programmed. Differently from apoptosis, the molecules involved in the necroptosis pathway could induce morphological features such as cell swelling, inflammatory responses, intracellular contents emission into the extracellular side, and dissipation of ions [26,45]. Subsequently, the discovery of the tumor necrosis factor (TNF) which induces necrosis, suggests otherwise. Indeed, the activation of specific death receptors or Toll-like receptors leads to the initiation of necroptosis [44]. The activation of death receptors, such as TNF alpha receptor 1, leads to the recruitment of proteins, including the cellular inhibitors of apoptosis 1 and 2 (c-IAP1/2) and RIP1, namely RIPK1, which promotes the activation of protein complex 1. Subsequently, RIP1 is translocated into the cytosol and interacts with RIP3 in the necrosome, which determines the initiation of necroptosis [46,47]. RIP3 can phosphorylate the lineage kinase-like pseudokinase (MLKL), the executor of the pathway that translocates to the cell membrane and leads to membrane rupture [48,49]. Therefore, detecting the protein interactions or protein levels of the RIP1-RIP3-MLKL axis is useful for identifying the existence of necroptosis. In HD, it is anticipated that targeting molecules involved in programmed cell death pathways could be an important way to better investigate necrotic mechanisms and also to find a therapeutic approach.

### 2.4. Autophagy

Autophagy is an evolutionary conserved lysosomal degradation pathway that is able to eliminate toxic components typical of neurodegenerative diseases, such as AD, HD, and PD. In pathological conditions, neurons tend to produce misfolded proteins which are not degraded by the usual cellular protein quality control. For this reason, autophagy is crucial in order to maintain neuronal proteostasis. In mammalians, autophagy distinction is based on cargo delivery to lysosomes. First, microautophagy is characterized by quick degradation due to the invagination of the lysosomal membrane. This mechanism is poorly understood, especially in neurons. Chaperone-mediated autophagy (CMA) requires the presence of a consensus pentapeptide sequence, LysPheGluArgGln in the substrate protein before lysosomal degradation [50]. On the other hand, the macroautophagy mechanism isolates the cargo creating a double membrane vesicle, called the autophagosome. This vesicle can be fused to lysosomes to degrade toxic components. In these terms, a recent study [49] showed transcription factor E-B (TFEB), a master regulator of the autophagy–lysosome pathway (macroautophagy), as a target of PGC-1alpha. TFEB can promote polyQ-Htt clearance, and its impaired expression alters its function in HD pathogenesis [51,52]. These studies identified a probable target for HD and other disorders caused by protein aggregates. Different to macroautophagy, which targets full-length polyQ-Htt, CMA selectively targets amino-terminal fragments of Htt. Considering that these fragments are very cytotoxic and abundant in HD brains, an important study demonstrated that manipulation of the CMA pathway greatly reduced inclusion formation and, at the same time, improved HD phenotypes in R6/2 mice [53,54]. Manipulating autophagic pathways could be considered a possible approach to the development of a therapy.

Recently, the compound Nilotinib (Tasigna™) an Abelson (Abl) inhibitor and autophagy modulator [55,56,57,58], has been described to interact with Beclin1 through a Bcl-2 homology 3 (BH3) domain [59]. Given the relationship between Abl and neurodegeneration, Abl inhibition with nilotinib is thought to be a perfect drug treatment. However, a recent study showed no changes in autophagy and aggregation levels and no behavioral alterations in R6/2 mice treated with nilotinib [59]. Nevertheless, a current clinical trial (NCT03764215) aims to evaluate whether nilotinib can have beneficial effects in patients with HD.

### 2.5. Ferroptosis

The term “ferroptosis” was coined in 2012 to describe a regulated form of iron-dependent, non-apoptotic oxidative cell death induced by erastin and RSL3 [60].

Ferroptosis is iron-dependent, and it is a form of cell death induced by lipid-reactive oxygen species (L-ROS) and plasma membrane polyunsaturated fatty acid depletion [61]. It has been observed that redox-active iron pools are able to directly catalyze the propagation of lipid peroxidation to form damaging species that lead to death [62].

This cell death is triggered by small molecules or conditions that inhibit glutathione (GSH) synthesis or the glutathione-dependent antioxidant enzyme glutathione peroxidase 4 (GPX4). In this way direct inhibition of synthesis should trigger ferroptosis [63,64].

Following the discovery of ferroptosis, the development of the first small-molecule inhibitor of ferroptosis, called ferrostatin-1, took place. Moreover, glutamate-induced ferroptosis in organotypic rat brain slices was demonstrated, suggesting the potential function of ferroptosis in neurodegeneration [65,66]. In fact, it has been speculated that ferroptosis may contribute to various diseases such as Alzheimer’s disease (AD), Parkinson’s disease (PD), but also to the toxic effect of mHTT causing HD. Specifically, HD has been linked to dysregulation of iron, glutamate, and glutathione. Further evidence that ferroptosis plays a crucial role in cell death in HD is the ability of the inhibitor ferrostatin-1 to abrogate mHTT-induced cell death [66]. A study on mouse HD models showed an accumulation of toxic iron in neurons of HD mice compared to the wild-type animals [67,68]. Magnetic resonance imaging (MRI) also demonstrated the presence of iron accumulation in brains of HD patients. This evidence suggests that iron accumulation may contribute to the neurodegenerative process in this disease [67,68]. Another study on mice with GPX4 ablation demonstrated the importance of inhibiting ferroptosis to prevent spinal motor neuron degeneration, because mice with this feature showed many motor disturbances [68]. In addition, it has been observed that a mHTT leads to increased oxidative stress, which consequently increases ROS levels in the cell [69]. Under normal conditions, GSH regulates GPX4 activity, and its function is to inhibit ferroptosis and eliminate excess lipid peroxides. However, an increase in ROS levels, and consequently an increase in lipid peroxides, leading to GSH depletion, decreases GPX4 [70]. In patients with HD, there is a deregulation of GSH that interferes with their functions and enzymes dependent on its action [70,71]. In patients with HD and asymptomatic carriers, high levels of lipid oxidation and low plasma levels of GSH demonstrate that oxidative stress may be related to the pathophysiological mechanism of HD [72]. Several research groups are trying to find therapies by targeting ferroptosis. Coenzyme Q10 (CoQ10) is one of the most promising candidates for ferroptosis inhibition because it is localized in the cellular and mitochondrial membranes. Considering that CoQ10 is the only endogenous antioxidant, in vivo experiments through an addition of Coenzyme Q10 showed a decreased lipids peroxidation and also a GSH upregulation [73,74]. Another kind of therapeutic approach could be the usage of iron chelation treatments such as Deferiprone and Deferasirox considering that they are able to successfully reduce lipid peroxidation [74].

## 3. NLRP3 Inflammasome/Pyroptosis Activation

Pyroptosis is a form of regulated cell death (RCD) described for the first time by Zychlinsky et al. in 1992 as apoptosis, but in 2001 the name “pyroptosis” was given by Cookson and Brennan to emphasize its inflammatory nature [75,76].

The pro-inflammatory nature of this programmed cell death differentiates it from other forms of RCD. Pyroptosis is the most immunogenic form of RCD of all cell death mechanisms and has well-defined morphological features such as cell swelling, chromatin condensation, and plasma membrane permeabilization [77,78].

Pyroptosis is mediated by several caspases: caspase-1, caspase-4, and caspase-5 in humans and caspase-11 in mice. Caspase-1 is activated following the activation of several inflammasomes, whereas caspase-4, caspase-5, and caspase-11 are activated after direct binding of LPS from gram-negative bacteria [79,80]. The majority of inflammasome sensors that have been identified contain a NOD-like receptor (NLR) domain, characterized by three distinct entities: a common NACHT domain; a leucine-rich repeats (LRRs) domain, and one or both death domains, PYD or CARD mediating ASC/caspase-1 interaction [81]. The first inflammasome to be identified was the NLRP1 (NOD-, LRR- and pyrin domain-containing 1) followed by the identification of additional NLR-containing inflammasomes, including NLRP3, NLRP6, NLRP7, NLRP12, and NLRC4 (NOD-, LRR- and CARD-containing 4) which are activated in response to a broad range of molecular signals. The inflammasome pathway can be canonical or non-canonical. The canonical pathway uses inflammasomes that activate caspase-1, whereas the non-canonical pathway activates other caspases. Activated caspase-1 is responsible for the cleavage of pro-IL-1β and pro-IL-18. These cytokines in their biologically active form will be ready to be released from host cells. In addition, caspase-1 also cleaves cytosolic gasdermine D (GSDMD) into an N-terminal fragment (GSDMD-N) of 242 amino acids and a C-terminal fragment (GSDMD-C) of 199 amino acids, in humans. GSDMD-N integrates into the membrane to form a pore that causes disruption of the cell membrane and release of cytokines (IL-1β and IL-18) and various cytosolic contents into the extracellular space increasing the inflammatory cascade in the tissue [81,82].

The non-canonical inflammasome pathway is initiated by the binding of lipopolysaccharide (LPS) from gram-negative bacteria directly onto caspase-4/5 in humans and caspase-11 in murine. Binding of LPS on these caspases promotes their oligomerization and activation. These caspases can cleave GSDMD to release GSDMD-N and trigger pyroptosis [80].

Inflammasome activation and pyroptosis were both found in many neurodegenerative diseases such as multiple sclerosis, stroke, ALS, HD and traumatic brain injury.

Our group described, for the first time, the involvement of pyroptosis in the pathogenesis of NLRP3 and caspase-1 activation in the mouse model of HD [83] (Figure 1).

## 4. Targeting the NLRP3 Inflammasome in HD

Recently, Jiang Jing and coworkers (2019) showed that Gal3 directly interacts with NLRP3 in activated macrophages and promotes the activation of NLRP3 inflammasome [84]. In this study, the authors demonstrated that suppression of Gal3 concurrently attenuates the activation of NLRP3 inflammasomes in HD microglia. In fact, authors reported, for the first time, the discovery of endogenous Gal3 puncta formation in microglia of neurodegenerative disease. More precisely, they showed that Gal3 puncta were colocalized with lysosomes (LAMP1 and LAMP2) in primary HD microglia that contain mHTT intracellularly. mHTT might trigger lysosome rupture, allowing Gal3 to bind to β-galactoside-containing glycoconjugates originally located in the luminal side of lysosomes and form puncta. This interferes with the clearance of damaged lysosomes and ultimately contributes to the over-activation of the neuroinflammatory response in HD microglia [84].

Moreover, Chen and coauthors (2019) demonstrated that HD patients had higher levels of plasma Gal3 than non-HD subjects. Importantly, the elevated plasma Gal3 levels are associated with the clinical features of HD. Similarto the findings in human plasma, plasma Gal3 levels in R6/2 mice were significantly upregulated at the disease manifested stage (12 weeks old). These findings revealed a previously unknown role of microglial Gal3 in HD pathogenesis and provide a new target for the development of novel therapeutic treatments. Specifically, the development of blood–brain barrier-permeable Gal3 inhibitors is a promising strategy worthy of further investigation.

The present study paved the way for the study of NLRP3 inflammasome/pyroptosis activation in HD.

Our group demonstrated that the expression of NLRP3 was significantly high in the 13-week-old R6/2 mouse striatum, where it was particularly distributed into the spiny projection neurons labeled by calbindin immunofluorescence. These are the neurons that degenerate massively in Huntington’s disease [85].

Furthermore, we observed that striatal interneurons that are typically resistant to neurodegeneration in HD such as cholinergic neurons, expressed low levels of NLRP3, confirming that they were excluded from the ongoing inflammasome process and related pyroptosis-associated striatal neuronal degeneration. Conversely, evidence of the inflammasome complex, which is part of the pyroptosis pathway, was observed in the parvalbuminergic and calretininergic striatal neurons, which are prone to degenerate in HD.

Moreover, our group showed that Olaparib, a PARP-1 inhibitor, was beneficial in reducing clinical manifestations of HD, that in the R6/2 model is expressed by clasping behavior and motor deficits during rotarod performance. Thus, we drew the inference that Olaparib is beneficial for HD symptoms. The most interesting aspect of this research was that Olaparib was able to reduce the expression of NLRP3 with subsequent inactivation of caspase-1 with a very specific modulation of the number and morphology of microglia cells [85]. The effects of Olaparib on microglial cells and on NLRP3 are, therefore, further proof of the interplay between inflammation and neurodegeneration in HD. The study of this compound, even though we had already shown beneficial effects of PARP inhibition in HD, was aimed at the possible repurposing of a drug that is already available, and therefore to expedite the bench-to-bedside process.

Kai-Po and coworkers (2021) recently demonstrated that long-term treatment of fully symptomatic HD mice with a selective NLRP3 inhibitor (MCC950) not only reduced the activation of NLRP3 and ROS production, but also rescued neuronal survival and attenuated gliosis in HD [86]. Oral administration of MCC950 halted the disease progression and markedly enhanced lifespan in a transgenic mouse model (R6/2) of HD through the inhibition of the NLRP3 inflammasome overactivation pathway [86].

## 5. Conclusions

Given the emerging role of NLRP3 inflammasome in the neurodegenerative process, which is shared by many pathological conditions, such as Alzheimer’s disease, Parkinson’s disease, and HD, the effort of many researchers in targeting NLRP3 or inflammasome components is quite coherent. In particular, HD is a genetic disease for which treating the genetic defect would be a very difficult task. Therefore, targeting specific markers, such as NLRP3, that can reduce the excessive inflammatory response, especially in the pre-symptomatic stage of the disease is considered a powerful approach. In fact, downmodulating neuroinflammation that is responsible for the neurodegenerative process and clinical symptoms represents an invaluable possibility to improve the quality of life not only for the fully symptomatic patients but, above all, for the pre-symptomatic generations.

## Figures and Tables

**Figure 1 ijms-23-08363-f001:**
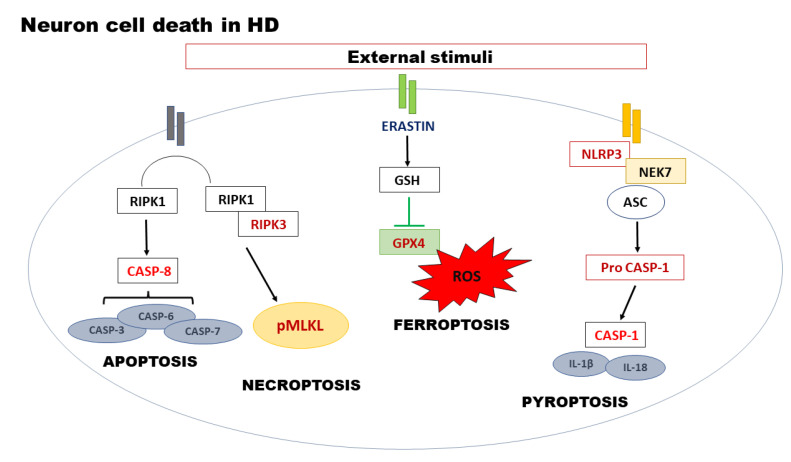
Schematic overview of cell death pathways. Neuron cell death is triggered by various external stimuli, which determine the activation of the different programmed cell death characterized by the activation of own executors.

## Data Availability

Not applicable.

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
