# Peer review of "Emerging Role of NLRP3 Inflammasome/Pyroptosis in Huntington’s Disease"

_ijms, 2022, doi:10.3390/ijms23158363_

Round 1

Reviewer 1 Report

In this manuscript, the authors review the role of the NLRP3 inflammasome in the pathogenesis of Huntington's disease, and discuss evidence that supports targeting the NLRP3 inflammasome therapeutically.

Overall I found this review article to be well put together and concisely discussed pathways of degeneration as they relate to Huntington's disease.

My main suggestion would be that the authors include a figure that visually summarizes the inflammatory/cell death molecular pathways discussed throughout the manuscript. This would greatly aid the reader in understanding the molecules and pathways as they are discussed.

Additionally, I detected a few minor grammatical errors in the manuscript that the authors should correct before publication. They are listed below:

Line 51-52: "the inflammatory process is set to initiate the healing mechanism thus protect the organism"

Line 53: "Activated microglia is able, to produce several dangerous.."

Line 56-57: "leading to the neurodegeneration"

Line 110: "many research groups focused on mitochondria role in disease progression"

Line 161-163: "During Chaperone-mediated Autophagy (CMA) before lysosomal degradation, requires the presence of a consensus pentapeptide sequence"

Line 169: "This kind of works identified a probable target"

Line 175: "The manipulation of these autophagic pathways could be considered candidate for therapy development."

Line 306-308: "Moreover, our group showed that Olaparib, a PARP-1 inhibitor, was beneficial in the clinical manifestation of HD disease, that in the R6/2 model express themselves as the clasping behavior and motor deficits during rotarod performance."

Author Response

My main suggestion would be that the authors include a figure that visually summarizes the inflammatory/cell death molecular pathways discussed throughout the manuscript. This would greatly aid the reader in understanding the molecules and pathways as they are discussed.

The figure that summarizes the inflammatory/cell death molecular pathways discussed has been included in the manuscript.

Additionally, I detected a few minor grammatical errors in the manuscript that the authors should correct before publication. They are listed below:

Line 51-52: "the inflammatory process is set to initiate the healing mechanism thus protect the organism"

Line 53: "Activated microglia is able, to produce several dangerous.."

Line 56-57: "leading to the neurodegeneration"

Line 110: "many research groups focused on mitochondria role in disease progression"

Line 161-163: "During Chaperone-mediated Autophagy (CMA) before lysosomal degradation, requires the presence of a consensus pentapeptide sequence"

Line 169: "This kind of works identified a probable target"

Line 175: "The manipulation of these autophagic pathways could be considered candidate for therapy development."

Line 306-308: "Moreover, our group showed that Olaparib, a PARP-1 inhibitor, was beneficial in the clinical manifestation of HD disease, that in the R6/2 model express themselves as the clasping behavior and motor deficits during rotarod performance." 

The suggested grammatical errors have been evaluated and modified accordingly

Reviewer 2 Report

In the manuscript "Emerging role of NLRP3 Inflammasome/Pyroptosis in Huntington's Disease" , the authors summarize the different cell death pathways in Huntington's disease (HD), mainly focusing on pyroptosis. They also focus on NLRPS3 activation and regulation in HD. Although the manuscript is well written and covers most of the important research findings, there are some minor things that will improve the review further for readers.

In the abstract and introduction, the authors wrote IT15 not italic, it should be italic like "IT15" also they wrote huntingtin protein as italic, this should be not italic.  

In part two, it will be better if they changed the title to "Neuronal cell death in Huntington's Disease since they didn't give information about mainly other diseases. 

In part 2.2 Apoptosis, in line 111, they mentioned increased mitochondrial membrane depolarization in HD cells. It will be better if they can specify the cell type such as striatal neuron etc.

In part two, they didn't provide enough study on the different cell death in HD, especially since there is no example for necroptosis part. They can add more information on the role of different cell death in HD. 

1)Vandenabeele, P., Galluzzi, L., Vanden Berghe, T. & Kroemer, G. Molecular mechanisms of necroptosis: an ordered cellular explosion. Nat. Rev. Mol. Cell Biol. 11, 700–714 (2010).

2)Hickey, Miriam A., and Marie-Françoise Chesselet. "Apoptosis in Huntington's disease." Progress in Neuro-Psychopharmacology and Biological Psychiatry 27.2 (2003): 255-265.

Author Response

Reviewer 2

In the abstract and introduction, the authors wrote IT15 not italic, it should be italic like "IT15" also they wrote huntingtin protein as italic, this should be not italic.  

Word changes have been done.

In part two, it will be better if they changed the title to "Neuronal cell death in Huntington's Disease since they didn't give information about mainly other diseases. 

The title has been changed.

In part 2.2 Apoptosis, in line 111, they mentioned increased mitochondrial membrane depolarization in HD cells. It will be better if they can specify the cell type such as striatal neuron etc.

In Line 111 the cell type has been specified.

In part two, they didn't provide enough study on the different cell death in HD, especially since there is no example for necroptosis part. They can add more information on the role of different cell death in HD. 

1)Vandenabeele, P., Galluzzi, L., Vanden Berghe, T. & Kroemer, G. Molecular mechanisms of necroptosis: an ordered cellular explosion. Nat. Rev. Mol. Cell Biol. 11, 700–714 (2010).

2)Hickey, Miriam A., and Marie-Françoise Chesselet. "Apoptosis in Huntington's disease." Progress in Neuro-Psychopharmacology and Biological Psychiatry 27.2 (2003): 255-265.

Suggested references have been included in the manuscript
